# Towards scalable and non-IID robust Hierarchical Federated Learning via Label-driven Knowledge Aggregator

## Abstract

In real-world applications, Federated Learning (FL) meets two challenges: (1) scalability, especially when applied to massive IoT networks, and (2) how to be robust against an environment with heterogeneous data. Realizing the first problem, we aim to design a novel FL framework named Full-stack FL (F2L). More specifically, F2L utilizes a hierarchical network architecture, making extending the FL network accessible without reconstructing the whole network system. Moreover, leveraging the advantages of hierarchical network design, we propose a new label-driven knowledge distillation (LKD) technique at the global server to address the second problem. As opposed to current knowledge distillation techniques, LKD is capable of training a student model, which consists of good knowledge from all teachers' models. Therefore, our proposed algorithm can effectively extract the knowledge of the regions' data distribution (i.e., the regional aggregated models) to reduce the divergence between clients' models when operating under the FL system with non-independent identically distributed data. Extensive experiment results reveal that: (i) our F2L method can significantly improve the overall FL efficiency in all global distillations, and (ii) F2L rapidly achieves convergence as global distillation stages occur instead of increasing on each communication cycle.

## 1 Introduction

Recently, Federated Learning (FL) is known as a novel distributed learning methodology for enhancing communication efficiency and ensuring privacy in traditional centralized one McMahan et al. (2017). However, the most challenge of this method for client models is non-independent and identically distributed (non-IID) data, which leads to divergence into unknown directions. Inspired by this, various works on handling non-IID were proposed in Li et al. (2020); Acar et al. (2021); Dinh et al. (2021a); Karimireddy et al. (2020); Wang et al. (2020); Zhu et al. (2021); Nguyen et al. (2022b). However, these works mainly rely on arbitrary configurations without thoroughly understanding the models' behaviors, yielding low-efficiency results. Aiming to fulfil this gap, in this work, we propose a new hierarchical FL framework using information theory by taking a deeper observation of the model's behaviors, and this framework can be realized for various FL systems with heterogeneous data. In addition, our proposed framework can trigger the FL system to be more scalable, controllable, and accessible through hierarchical architecture. Historically, anytime a new segment (i.e., a new group of clients) is integrated into the FL network, the entire network must be retrained from the beginning. Nevertheless, with the assistance of LKD, the knowledge is progressively transferred during the training process without information loss owing to the empirical gradients towards the newly participated clients' dataset.

The main contributions of the paper are summarized as follows. **(1)** We show that conventional FLs performance is unstable in heterogeneous environments due to non-IID and unbalanced data by carefully analyzing the basics of Stochastic Gradient Descent (SGD). **(2)** We propose a new multi-teacher distillation model, Label-Driven Knowledge Distillation (LKD), where teachers can only share the most certain of their knowledge. In this way, the student model can absorb the most meaningful information from each teacher. **(3)** To trigger the scalability and robustness against non-IID data in FL, we propose a new hierarchical FL framework, subbed Full-stack Federated Learning (F2L). Moreover, to guarantee the computation cost at the global server, F2L architecture

integrates both techniques: LKD and FedAvg aggregators at the global server. To this end, our framework can do robust training by LKD when the FL process is divergent (i.e., at the start of the training process). When the training starts to achieve stable convergence, FedAvg is utilized to reduce the server's computational cost while retaining the FL performance. **(4)** We theoretically investigate our LKD technique to make a brief comparison in terms of performance with the conventional Multi-teacher knowledge distillation (MTKD), and in-theory show that our new technique always achieves better performance than MTKD. **(5)** We validate the practicability of the proposed LKD and F2L via various experiments based on different datasets and network settings. To show the efficiency of F2L in dealing with non-IID and unbalanced data, we provide a performance comparison and the results show that the proposed F2L architecture outperforms the existing FL methods. Especially, our approach achieves comparable accuracy when compared with FedAvg (McMahan et al. (2017)) and higher $7 - 20\%$ in non-IID settings.

## 2 RELATED WORK

### 2.1 FEDERATED LEARNING ON NON-IID DATA

To narrow the effects of divergence weights, some recent studies focused on gradient regularization aspects Li et al. (2020); Acar et al. (2021); Dinh et al. (2021a); Karimireddy et al. (2020); Wang et al. (2020); Zhu et al. (2021); Nguyen et al. (2022b). By using the same conceptual regularization, the authors in Li et al. (2020); Acar et al. (2021), and Dinh et al. (2021a) introduced the FedProx, FedDyne, and FedU, respectively, where FedProx and FedDyne focused on pulling clients' models back to the nearest aggregation model while FedU's attempted to pull distributed clients together. To direct the updated routing of the client model close to the ideal server route, the authors in Karimireddy et al. (2020) proposed SCAFFOLD by adding a control variate to the model updates. Meanwhile, to prevent the aggregated model from following highly biased models, the authors in Wang et al. (2020) rolled out FedNova by adding gradient scaling terms to the model update function. Similar to Dinh et al. (2021a), the authors in Nguyen et al. (2022b) launched the WALF by applying Wasserstein metric to reduce the distances between local and global data distributions. However, all these methods are limited in providing technical characteristics. For example, Wang et al. (2020) demonstrated that FedProx and FedDyne are ineffective in many cases when using pullback to the globally aggregated model. Meanwhile, FedU and WAFL have the same limitation on making a huge communication burden. Aside from that, FedU also faces a very complex and non-convex optimization problem.

Regarding the aspect of knowledge distillation for FL, only the work in Zhu et al. (2021) proposed a new generative model of local users as an alternative data augmentation technique for FL. However, the majority drawback of this model is that the training process at the server demands a huge data collection from all users, leading to ineffective communication.

Motivated by this, we propose a new FL architecture that is expected to be more elegant, easier to implement, and much more straightforward. Unlike Dinh et al. (2021a); Acar et al. (2021); Karimireddy et al. (2020), we utilize the knowledge from clients' models to extract good knowledge for the aggregation model instead of using model parameters to reduce the physical distance between distributed models. Following that, our proposed framework can flexibly handle weight distance and probability distance in an efficient way, i.e., $\|p^k(y = c) - p(y = c)\|$ (please refer to Appendix B).

### 2.2 MULTI-TEACHER KNOWLEDGE DISTILLATION

MTKD is an improved version of KD (which is presented in Appendix A.2), in which multiple teachers work cooperatively to build a student model. As shown in Zhu et al. (2018), every MTKD technique solves the following problem formulation:

$$\textbf{P1} : \min \mathcal{L}_m^{KL} = \sum_{r=1}^{R} \sum_{l=1}^{C} \hat{p}(l|\boldsymbol{X}, \boldsymbol{\omega}^r, T) \log \frac{\hat{p}(l|\boldsymbol{X}, \boldsymbol{\omega}^r, T)}{\hat{p}(l|\boldsymbol{X}, \boldsymbol{\omega}^g, T)}, \tag{1}$$

here, $r \in \{R\}$ are the teachers' indices. By minimizing **P1**, the student $\hat{p}^g$ can attain knowledge from all teachers. However, when using MTKD, there are some problems in extracting the knowledge distillation from multiple teachers. In particular, the process of distilling knowledge in MTKD is typically empirical without understanding the teacher's knowledge (i.e., aggregating all KL divergences

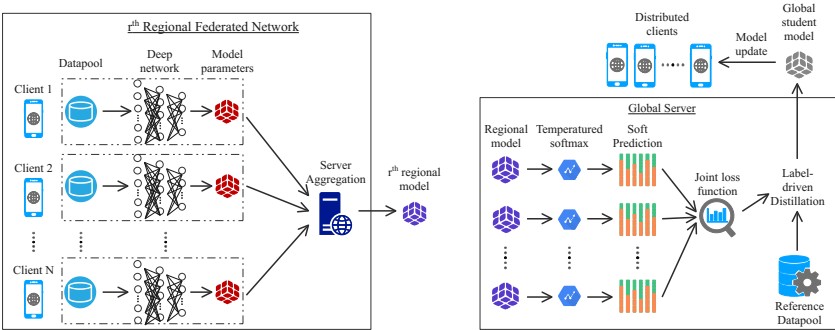

Figure 1: The architecture of our F2L framework.

between each teacher and the student). Therefore, MTKD is unable to exploit teachers' detailed predictions for the KD (e.g., Liu et al. (2020c), Asif et al. (2019), Zhu et al. (2018), Fukuda et al. (2017), Tran et al. (2020)). Another version of MTKD, KTMDs can only apply for a better teachers to distill knowledge (e.g., Shen et al. (2019), Zhu & Wang (2021), Zhang et al. (2022), Son et al. (2021)). For example, as provided in (Shen et al., 2019, eq. 6), the student only selects the best teacher to operate the knowledge distillation. Visually, this technique is the same as the way of selecting a teacher among a set of teachers to carry out a single teacher distillation. Therefore, the student's performance is always bounded by the best teacher's performance. Another popular research direction in MTKD is to leverage the advantage of the gap between teachers' hidden class features. However, owing to the lack of explanatory knowledge in teachers' hidden layers, the method in Zhu & Wang (2021) cannot obtain better student performance when compared to their teachers. Generally, current MTKD techniques cannot extract good knowledge from different customer models, leading to weight divergence in FL.

## 3 FULL-STACK FEDERATED LEARNING

### 3.1 THE F2L FRAMEWORK

The main objective of our work is to design a hierarchical FL framework, in which a global server manages a set of distinct regional servers. Utilizing hierarchical FL, our proposed algorithm can achieve computation and computation efficiency. The reason is that Hierarchical FL makes the clients to train sectionally before making the global aggregation Liu et al. (2020a); Briggs et al. (2020). Consequently, FL inherits every advantage from mobile edge intelligence concept over traditional non-hierarchical networks (e.g., communication efficiency, scalability, controlability) Pham et al. (2020); Luong et al. (2016); Lim et al. (2020). At the end of each knowledge-sharing episode, the regions (which are supervised by regional servers) cooperate and share their knowledge (each region functions as a distinguished FL system, with a set amount of communication rounds per episode).

In each episode, each region randomly selects a group of clients from the network to carry out the aggregation process (e.g., FedAvg, FedProx); therefore, each region functions as a small-scale FL network. As a result, there are always biases in label-driven performance by applying random sampling on users per episode (see Appendix F). Given the random sampling technique applied to the regional client set, the regions always have different regional data distributions. Consequently, various label-driven performances of different teachers might be achieved.

At the global server, our goal is to extract good performance from regional teachers while maintaining the salient features (e.g., understanding of the regional data distributions) of all regions. As a result, we can capture useful knowledge from diverse regions in each episode using our proposed innovative knowledge distillation technique (which is briefly demonstrated in Section 3.2). We train the model on the standard dataset on the central server to extract knowledge from multiple teachers into the global student model. The preset data pool on the server $\mathcal{S}$ is used to verify the model-class reliability and generate pseudo labels.

The system model is illustrated in Fig. 1, and thoroughly described in Appendix C. The pseudo

algorithm for F2L is demonstrated in Algorithm 1. When the FL process suffers from client-drift Karimireddy et al. (2020) (i.e., the distribution of label-driven accuracies of different regions have large variance), the F2L framework applies LKD to reduce the class-wise performance gaps between regions (i.e., the regions with better performance on a particular class share their knowledge to regions with low performance). As a result, the FL network achieves a significantly faster convergence when utilizing LKD (which is intensively explained in Section 3.2.) for the global aggregation process. When the generalization gap between regions is considerably reduced (i.e., $\| \max_r \beta_r^c - \min_r \beta_r^c \| \leq \epsilon$), our F2L network becomes vanilla FL to reduce computation and communication costs. To this end, our method can achieve computation efficiency while showing robustness to the non-IID data in the network. Additionally, whenever a new set of clients are added into the network and makes positive contributions to the FL system (e.g., $\| \max_r \beta_r^c - \min_r \beta_r^c \| \geq \epsilon$ where $\| \max_r \beta_r^c \|$ a corresponding to the new region's performance) the LKD aggregator can be switched back to improve the FL system's performance over again.

## 3.2 LABEL-DRIVEN KNOWLEDGE DISTILLATION

To extract knowledge from multiple teachers to the global student model, we train the model on the standard dataset on the central server. The preset data pool on the server $\mathcal{S}$ is used to verify the model-class reliability and generate pseudo labels. In our work, the MTKD process executes two main tasks: (1) extracting the teachers' knowledge and (2) maintaining the students' previous performance.

To comprehend the LKD method, we first revisit the conventional MTKD, where the probabilistic output is calculated by model $\boldsymbol{\omega}$ on $x_i$, the prediction label $c$ is $\hat{p}(l|x_i, \boldsymbol{\omega}, T, c)$ and its relation is:

$$\hat{p}(l|x_i, \boldsymbol{\omega}, T, c) = \begin{cases} \hat{p}(l|x_i, \boldsymbol{\omega}, T), & \text{if argmax} [\hat{p}(l|x_i, \boldsymbol{\omega}, T)] = c, \\ 0, & \text{otherwise.} \end{cases} \tag{2}$$

On the one hand, we aim to transfer knowledge from different regional models to the global one. Inspired by Hinton et al. (2015), we use the Kullback–Leibler (KL) divergence between each regional teacher and the global logits as a method to estimate the difference between two models' performance. The relationship is expressed as follows:

$$\mathcal{L}_r^{KL} = \sum_{c=1}^{C} \beta_r^c \sum_{i=1}^{S_c^r} \sum_{l=1}^{C} \hat{p}^r(l|x_i, \boldsymbol{\omega}^r, T, c) \times \log \frac{\hat{p}^r(l|x_i, \boldsymbol{\omega}^r, T, c)}{\hat{p}^g(l|x_i, \boldsymbol{\omega}^g, T, c)}, \tag{3}$$

where $S$ is the number of samples of the fixed dataset $\mathcal{S}$ on the server. $(\boldsymbol{X}_{\text{alg}}^r, \boldsymbol{Y}_{\text{alg}}^r)$ is the dataset which is pseudo labeled and aligned by regional model $r$ and $(\boldsymbol{X}_{\text{alg}}^r[c], \boldsymbol{Y}_{\text{alg}}^r[c])$ represents the set of data with size of $S_c^r$ labeled by the model $r$ as $c$. Although the same preset dataset is utilized on every teacher model, the different pseudo labeling judgments from different teachers lead to the different dataset tuples. The process of identifying $S_c^r$ is demonstrated in Algorithm 3. Because the regional models label on the same dataset $S$, we have $\sum_{c=1}^{C} S_c^r = S$ for all regional models. $D_{KL}^c(\hat{p}^r || \hat{p}^g)$ is the $c$ label-driven KL divergence between model $r$ and model $g$.

On the other hand, we aim to guarantee that the updated global model does not forget the crucial characteristics of the old global model. Hence, to measure the divergence between the old and the updated model, we introduce the following equation:

$$\mathcal{L}_{\boldsymbol{\omega}_{upd}}^{KL} = \sum_{c=1}^{C} \beta_{\boldsymbol{\omega}_{old}}^c \sum_{i=1}^{S_c^r} \sum_{l=1}^{C} \hat{p}^g(l|x_i, \boldsymbol{\omega}_{old}^g, T, c) \times \log \frac{\hat{p}^g(l|x_i, \boldsymbol{\omega}_{old}^g, T, c)}{\hat{p}^g(l|x_i, \boldsymbol{\omega}_{new}^g, T, c)}, \tag{4}$$

where $\boldsymbol{\omega}_{old}$ is the old parameters set of the global model which is distilled in the last episode of F2L. More details about the label-driven knowledge distillation are discussed in Appendix G.

To compare the performance between LKD and MTKD, we consider the following assumption and lemmas:

**Lemma 1** *Given $\tau_r^c$ is the c-label driven predicting accuracy on model $r$. Let $\sigma_{r,c}^2, \mu_{r,c}$ be the model's variance and mean, respectively. The optimal value of variance and mean on student model (i) $\sigma_{LKD,g,c}^{*2}, \mu_{LKD,g,c}^*$ yields $\sigma_{LKD,g,c}^{*2} = \frac{1}{\sum_{r=1}^{R} e^{\tau_r^c}} \sum_{r=1}^{R} e^{\tau_r^c} \sigma_{r,c}^2$, and $\mu_{LKD,g,c}^* = \frac{1}{\sum_{r=1}^{R} e^{\tau_r^c}} \sum_{r=1}^{R} e^{\tau_r^c} \mu_{r,c}.$.*

*Proof:* The proof is provided in Appendix J.

**Assumption 1** *Without loss of generality, we consider $R$ distinct regional models whose accuracy satisfy the following prerequisites $\sigma_{1,c}^2 \leq \sigma_{2,c}^2 \leq \ldots \leq \sigma_{R,c}^2$, and $|\mu_{1,c} - \bar{\mu}_c| \leq |\mu_{2,c} - \bar{\mu}_c| \leq \ldots \leq |\mu_{R,c} - \bar{\mu}_c|$ ($\bar{\mu}_c$ is denoted as an empirical global mean of the dataset on class $c$).*

**Lemma 2** *Given the set of models with variance satisfy $\sigma_{1,c}^2 \leq \sigma_{2,c}^2 \leq \ldots \leq \sigma_{R,c}^2$, the models' accuracy have the following relationship $\tau_1^c \geq \tau_2^c \geq \ldots \geq \tau_R^c$.*

*Proof.* The proof can be found in Appendix K.

**Theorem 1** *Let $\sigma_{LKD,g,c}^{*2}$ be the class-wise variance of the student model, and $\sigma_{MTKD,g,c}^{*2}$ be the class-wise variance of the model of teacher $r$, respectively. We always have the student's variance using LKD technique always lower than that using MTKD:*

$$\sigma_{LKD,g,c}^{*2} \leq \sigma_{MTKD,g,c}^{*2}. \tag{5}$$

*Proof*: For the complete proof see Appendix H.

**Theorem 2** *Let $\mu_{LKD,g,c}^*$ be the empirical $c$-class-wise mean of the student model, and $\mu_{MTKD,g,c}^*$ be the empirical $c$-class-wise mean of the model of teacher $r$, respectively. We always have the student's empirical mean using LKD technique always closer to the empirical global dataset's class-wise mean ($\bar{\mu}_c$) than that using MTKD:*

$$|\mu_{LKD,g,c}^* - \bar{\mu}_c| \leq |\mu_{MTKD,g,c}^* - \bar{\mu}_c|. \tag{6}$$

Given Theorems 1 and 2, we can prove that our proposed LKD technique can consistently achieve better performance than that of the conventional MTKD technique. Moreover, by choosing the appropriate LKD allocation weights, we can further improve the LKD performance over MTKD. Due to space limitation, we defer the proof to Appendix I.

### 3.3 CLASS RELIABILITY SCORING

The main idea of class reliability variables $\beta_r^c$, $\beta_{\boldsymbol{\omega}_{old}}^c$ in LKD is to weigh the critical intensity of the specific model. Therefore, we leverage the attention design from Vaswani et al. (2017) to improve the performance analysis of teachers' label-driven.

For regional models with disequilibrium or non-IID data, the teachers only teach the predictions relying upon their specialization. The prediction's reliability can be estimated by leveraging the validation dataset on the server and using the function under the curve (AUC) as follows:

$$\beta_r^c = \frac{\exp(f_{AUC}^{c,r} T_{\boldsymbol{\omega}})}{\sum_{r=1}^R \exp(f_{AUC}^{c,r} T_{\boldsymbol{\omega}})}, \tag{7}$$

where $f_{AUC}^{c,r}$ denotes the AUC function on classifier $c$ of the regional model $r$. Since AUC provides the certainty that a specific classifier can work on a label over the rest, we use the surrogate softmax function to weigh the co-reliability among the same labeling classifiers on different teacher models. For simplicity, we denote $\beta_{\boldsymbol{\omega}_{old}}^c$ as the AUC on each labeling classifier:

$$\beta_{\boldsymbol{\omega}_{old}}^c = \frac{\exp(f_{AUC}^{c,\boldsymbol{\omega}_{old}} T_{\boldsymbol{\omega}})}{\exp(f_{AUC}^{c,\boldsymbol{\omega}_{new}} T_{\boldsymbol{\omega}}) + \exp(f_{AUC}^{c,\boldsymbol{\omega}_{old}} T_{\boldsymbol{\omega}})}. \tag{8}$$

In the model update class reliability, instead of calculating the co-reliability between teachers, equation 8 compares the performance of the previous and current global models. Moreover, we introduce a temperatured value for the class reliability scoring function, denoted as $T_{\boldsymbol{\omega}}$. By applying a large temperatured value, the class reliability variable sets $\beta_r^c$, and $\beta_{\boldsymbol{\omega}_{old}}^c$ make a higher priority on the better performance (i.e., the label-driven performance on class $c$ from teacher $r$, e.g., $f_{AUC}^{c,r}$ in equation equation 7 or class $c$ from old model $\boldsymbol{\omega}_{old}$ in equation equation 8). By this way, we can preserve the useful knowledge which is likely ignored in the new distillation episode. The more detailed descriptions of class reliability scoring are demonstrated in Algorithm 6.

### 3.4 JOINT MULTI-TEACHER DISTILLATION FOR F2L

We obtain the overall loss function for online distillation training by the proposed F2L:

$$\mathcal{L}_{\text{F2L}} = \lambda_1 \sum_{r=1}^{R} \mathcal{L}_r^{KL} + \lambda_2 \mathcal{L}_{\boldsymbol{\omega}_{upd}}^{KL} + \lambda_3 \mathcal{L}_{CE}^g, \tag{9}$$

where $\lambda_1, \lambda_2, \lambda_3$ are the scaling coefficients of the three terms in the joint loss function. The first and second terms imply the joint LKD from the regional teacher models and the updating correction step, respectively. Moreover, to ensure that knowledge the student receives from teachers is accurate and can be predicted accurately in practice, we need to validate the quality of the student model on the real data. Thus, we also compute the "standard" loss between the student and the ground-truth labels of the train dataset. This part is also known as the hard estimator, which is different from the aforementioned soft-distillator. The hard loss equation is as follows:

$$\mathcal{L}_{CE}^g = H(y, \hat{p}(l|\boldsymbol{X}, \boldsymbol{\omega}^g, T)) = \sum_{l=1}^{C} y_l \log \hat{p}(l|\boldsymbol{X}, \boldsymbol{\omega}^g, T). \tag{10}$$

---

**Algorithm 1** F2L framework

---

**Require:** Initialize clients' weights, global aggregation round, number of regions $R$, arbitrary $\epsilon$.
**while** not converge **do**
    **for** all regions $r \in \{1, 2, \ldots, R\}$ **do**
        **for** all user in regions **do**
            Apply FedAvg on regions $r$.
        **end for**
        Send regional model $\boldsymbol{\omega}^r$ to the global server.
    **end for**
    **if** reach global aggregation round **then**
        **if** $\| \max_r \beta_r^c - \min_r \beta_r^c \| \geq \epsilon$ where $\beta = \{\beta_r^1, \ldots, \beta_r^C\}|_{r=1}^R$ from Algorithm 6 **then**
            Apply LKD as described in Algorithm 2
        **else**
            $\boldsymbol{\omega}^g = 1/R \sum_{r=1}^R \boldsymbol{\omega}^r$.
        **end if**
    **end if**
**end while**

---

We use the temperature coefficient $T = 1$ to calculate the class probability for this hard loss. The overall training algorithm for LKD is illustrated in Algorithm 2. In terms of value selection for scaling coefficients, the old global model can be considered as an additional regional teacher's model in the same manner, in theory. Therefore, $\lambda_2$ should be chosen as:

$$\lambda_2 = \begin{cases} \frac{1}{R}\lambda_1, & \text{if update distillation in equation 4 is considered,} \\ 0, & \text{otherwise,} \end{cases} \tag{11}$$

where $R$ is the number of regions decided by our hierarchical FL settings. With respect to $\lambda_3$, the value is always set as:

$$\lambda_3 = \begin{cases} 1 - \frac{R+1}{R}\lambda_1, & \text{if update distillation in equation 4 is considered,} \\ 1 - \lambda_1, & \text{otherwise.} \end{cases} \tag{12}$$

### 3.5 DISCUSSIONS ON THE EXTENT OF PROTECTING PRIVACY

In its simplest version, our proposed F2L framework, like the majority of existing FL approaches, necessitates the exchange of models between the server and each client, which may result in privacy leakage due to, for example, memorization present in the models. Several existing protection methods can be added to our system in order to safeguard clients against enemies. These include adding differential privacy Geyer et al. (2017) to client models and executing hierarchical and decentralized model fusion by synchronizing locally inferred logits, for example on random public data, as in work Chang et al. (2019). We reserve further research on this topic for the future.

Table 1: The top-1 test accuracy of different baselines on different data settings. The $\alpha$ indicates the non-IID degree of the dataset (the lower value of $\alpha$ means that the data is more heterogeneous).

| Dataset | FedAvg | FedGen | FedProx | Fed-Distill | F2L (Ours) | FedAvg | FedGen | FedProx | Fed-Distill | F2L (Ours) |
|---|---|---|---|---|---|---|---|---|---|---|
| | Dirichlet ($\alpha = 1$) | | | | | Dirichlet ($\alpha = 0.1$) | | | | |
| EMNIST | 71.66 | 78.70 | 70.77 | 75.56 | **81.14** | 59.10 | 68.24 | 58.88 | 46.03 | **68.31** |
| CIFAR-10 | 60.48 | 59.21 | 63.72 | 62.36 | **71.22** | 47.07 | 47.08 | 47.05 | 45.67 | **55.22** |
| CIFAR-100 | 36.17 | 40.26 | 36.3 | 34.88 | **50.33** | 21.31 | 28.96 | 20.43 | 16.15 | **31.07** |
| CINIC-10 | 65.23 | 71.61 | 65.15 | 67.77 | **74.85** | 47.55 | 52.35 | 48.2 | 47.1 | **57.12** |
| CelebA | 70.82 | 75.43 | 71.07 | 68.59 | **81.65** | 63.58 | 70.14 | 66.33 | 62.91 | **74.14** |

## 4 EXPERIMENTAL EVALUATION

### 4.1 COMPARISON WITH FL METHODS

We run the baselines (see Section E) and compare with our F2L. Then, we evaluate the comparisons under different non-IID ratio. More precisely, we generate the IID data and non-IID data with two different Dirichlet balance ratio: $\alpha = \{1, 10\}$. The comparison results are presented in Table 1. As shown in Table 1, the F2L can outperform the four baselines with a significant increase in accuracy. The reason for this phenomenon is that the LKD technique selectively draws the good features from regional models to build a global model. Hence, the global model predicts the better result on each different class and the entire accuracy of the global model then increases tremendously. The significant impact when applying LKD to distill different teachers to one student is shown in Table 2.

### 4.2 COMPUTATION EFFICIENCY OF F2L

To evaluate the computation efficiency of our proposed F2L process, we compare our F2L process with 3 benchmarks: (i) F2L-noFedAvg (aggregator only consists of LKD), (ii) vanilla FL (FL with flatten architecture and FedAvg as an aggregator), and (iii) flatten LKD (FL with flatten architecture based with LKD as an alternate aggregator). Fig. 2(a) shows that the F2L system can achieve convergence as good as the F2L-noFedAvg. The reason is that: after several communication rounds, the distributional distance between regions is reduced thanks to the LKD technique. Hence, the efficiency of the LKD technique on the data is decreased. As a consequence, the LKD technique shows no significant robustness over FedAvg aggregator. In the non-hierarchical settings, the flatten LKD and FedAvg reveal under-performed compared to the proposed hierarchical settings. We assume that the underperformance above comes from the data shortage of clients' training models. To be more detailed, the clients' dataset are considerably smaller than that of the "regional dataset". Thus, the regional models contain more information than the clients' models. We believe that: in the LKD technique, teachers' models require a certain amount of knowledge to properly train a good student (i.e., the global model). Given the convergence rate from Fig. 2(a) and the computation cost at the

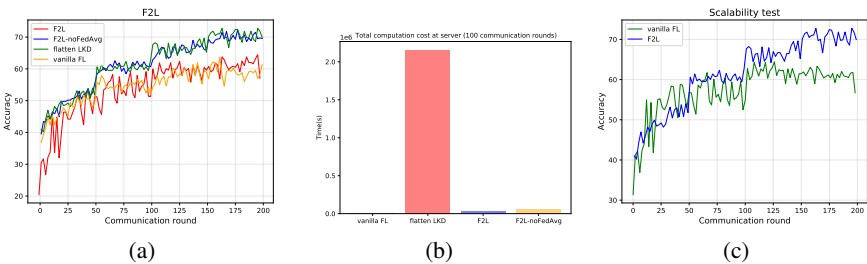

Figure 2: Performance benchmarks of F2L under different settings. Fig. 2(a) reveals the convergence. Fig. 2(b) shows the computational cost, and Fig. 2(c) demonstrates the F2L convergence when a new set of clients are added into the FL system (i.e., at communication round 100).

Table 2: Top-1 accuracy of F2L on 5 datasets MNIST, EMNIST, CIFAR-100, , CINIC-10 and CelebA. The data's heterogeneity is set at $\alpha = 0.1$ on CIFAR-100, MNIST, CINIC-10 and CelebA. We use EMNIST "unbalanced" to evaluate in this test. The "before update" and "after update" denote the teacher models' accuracies before and after the global distillation, respectively.

| | MNIST | | EMNIST | | CIFAR-100 | | CINIC-10 | | Celeb-A | |
|---|---|---|---|---|---|---|---|---|---|---|
| | before update | after update | before update | after update | before update | after update | before update | after update | before update | after update |
| Teacher 1 | 61.02 | **95.19** | 73.27 | **84.09** | 20.11 | **35.41** | 43.8 | **46.59** | 62.37 | **67.98** |
| Teacher 2 | 92.49 | **98.22** | 78.80 | **83.62** | 18.82 | **31.2** | 42.15 | **46.01** | 63.79 | **72.33** |
| Teacher 3 | 81.60 | **97.63** | 80.5 | **84.10** | 22.40 | **34.93** | 40.02 | **42.15** | 64.05 | **69.44** |
| G-student | **98.71** | | **84.11** | | **37.68** | | **47.65** | | **70.12** | |

server on Fig. 2(b), we can see that, by using the adaptive switch between LKD and FedAvg in F2L, we can achieve significant computational efficiency at the aggregation server. Note that F2L can substantially increase performance and computation efficiency compared with non-hierarchical architecture.

## 4.3 SCALABILITY

This section evaluates the F2L scalability. To do so, we inject a group of clients with non-IID data into our FL system after 100 rounds (when the convergence becomes stable). Note that the FL system has never trained these data. The detailed configurations of our experimental assessments can be found in Appendix E. As it can be seen from Fig. 2(c), when a new group of clients are added to the FL system, the vanilla FL shows a significant drop in terms of convergence. The reason is because of the distribution gap between the global model's knowledge and knowledge of the clients' data. Whenever new data with unlearned distribution is added to a stable model, the model will make considerable gradient steps towards the new data distribution. Thus, the FedAvg takes considerable learning steps to become stable again. In contrast, in F2L system, the learning from newly injected regions does not directly affect the learning of the whole FL system. Instead, the knowledge from the new domains is selectively chosen via the LKD approach. Thus, the LKD process does not suffer from information loss when new clients with non-IID data are added to the FL system.

## 4.4 LKD ANALYSIS

In this section, we evaluate the LKD under various settings to justify the capability of LKD to educate the good student from the normal teachers. Our evaluations are as follows.

**Can student outperform teachers?** To verify the efficiency of LKD with respect to enhancing student performance, we first evaluate F2L on MNIST, EMNIST, CIFAR-100, CINIC-10, CelebA dataset. The regions are randomly sampled from the massive FL network. In this way, we only evaluate the performance of LKD on random teachers. Table 2 shows top-1 accuracy on the regional teacher and student models. The results reveal that LKD can significantly increase the global model performance compared with that of the regional models. Moreover, the newly distilled model can work well under each regional non-IID data after applying the model update.

To make a better visualization for the LKD's performance, we reveal the result of LKD on EMNIST dataset in terms of confusion matrix as in Fig. 3. As it can be seen from the figure, the true predictions is represented by a diagonals of the matrices. A LKD performance is assumed to be well predicted when the value on diagonals is high (i.e., the diagonals' colors is darker), and the off-diagonals is low (i.e., the off-diagonals' colors are lighter). As we can see from the four figures, there are a significant reduce in the off-diagonals' darkness in the student performance (i.e., Fig. 3(d)) comparing to the results in other teachers (i.e., Figures 3(a), 3(b), and 3(c)). Therefore, we can conclude that our proposed MTKD techniques can surpass the teachers' performance as we mentioned in Section 2.

**Teachers can really educate student?** We evaluate LKD under different soft-loss coefficients $\lambda_1$ while the hard-loss factor is set at $\lambda_3 = 1 - \lambda_1$ (the scaling value $\lambda_2$ is set to 0). Thus, we can justify whether the robust performance of LKD comes from the joint distillation from teachers or just the exploitation of data-on-server training. We evaluate LKD on six scaling values

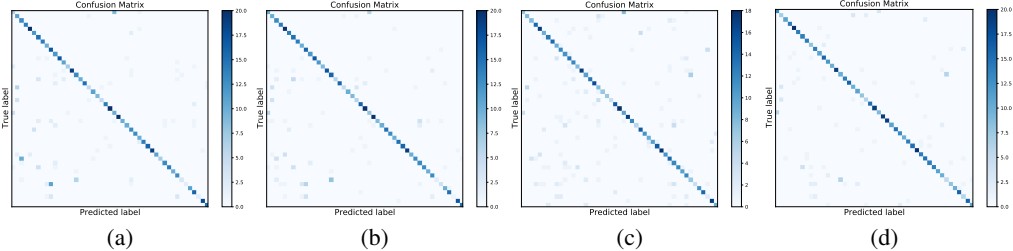

Figure 3: The illustrative results of LKD on EMNIST dataset. Confusion matrices show the effectiveness of joint distillation on regional models. Figures (a), (b), and (c) are the confusion matrix before distillation of teacher's predictions in region 1, 2, and 3, respectively (see Appendix E). Fig. (d) is the confusion matrix of predictions after distillation of student. The matrix diagonal demonstrates the true-predicted label of the model.

$\lambda_1 = \{0, 0.001, 0.01, 0.1, 0.5, 1\}$. We evaluate on three dataset, including EMNIST, CIFAR-10, and CIFAR-100, and summarize the results in Tables 5, 6 and 7 in **Appendices**. We can see from the three tables that the LKD cap off with $\lambda_3 = 0.01$. Especially, when $\lambda_3 = 1$ (which means the LKD acts as a vanilla cross-entropy optimizer), the model accuracy reduces notably. This means that the LKD only uses hard-loss as a backing force to aid the distillation. Thus, our LKD is appropriate and technically implemented.

**Required training sample size for joint distillation.** To justify the ability of LKD under a shortage of training data, we evaluate LKD with six different data-on-server settings: $\sigma = \{1, 1/2, 1/4, 1/6, 1/8, 1/10\}$, where $\sigma$ is the sample ratio when compared with the original data-on-server as demonstrated in Table 4. As we can see from the implementation results in three Tables 8, 9, and 10 in **Appendices**, the F2L is demonstrated to perform well under a relatively small data-on-server. To be more specific, we only need the data-on-server to be 4 times lower than the average data-on-client to achieve a robust performance compared with the vanilla FedAvg. However, we suggest using the data-on-server to be larger than the data from distributed clients to earn the highest performance for LKD. Moreover, due to the ability to work under unlabeled data, the data-on-server does not need to be all labeled. We only need a small amount of labeled data to aid the hard-loss optimizer. Thus, the distillation data on the server can be updated from distributed clients gradually.

## BROADER IMPACT AND LIMITATION

Due to the hierarchical framework of our proposed F2L, each sub-region acts like an independent FL process. Therefore, our F2L is integrable with other current methods, which means that we can apply varying FL techniques (e.g., FedProx, FedDyne, FedNova, HCFL Nguyen et al. (2022a)) into distinct sub-regions to enhance the overall F2L framework. Therefore, architecture search (e.g., which FL technique is suitable for distinct sub FL region) for the entire hierarchical network is essential for our proposed framework, which is the potential research for the future work. Moreover, the hierarchical framework remains unearthed. Therefore, a potentially huge amount of research directions is expected to be investigated (e.g., resource allocation Nguyen et al. (2022c); Saputra et al. (2022; 2021); Dinh et al. (2021b), and task offloading in hierarchical FL Yang et al. (2021)). However, our LKD technique still lacks of understanding about the teachers' models (e.g., how classification boundaries on each layer impact on the entire teachers' performance). By investigating in explainable AI, along with layer-wise performance, we can enhance the LKD, along with reducing the unlabeled data requirements for the distillation process in the future work.

## 5 CONCLUSION

In this research, we have proposed an FL technique that enables knowledge distillation to extract the-good-feature-only from clients to the global model. Our model is capable of tackling the FL's heterogeneity efficiently. Moreover, experimental evaluations have revealed that our F2L model outperforms all of the state-of-the-art FL baselines in recent years.

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
