# OpenReview forum: "Towards scalable and non-IID robust Hierarchical Federated Learning via Label-driven Knowledge Aggregator"
_ICLR.cc/2023/Conference — Submitted to ICLR 2023_

### Official Review · Reviewer_jW2m · 2022-10-20

**Confidence:** 4
**Clarity, Quality, Novelty And Reproducibility:** See the Strength And Weaknesses.
**Correctness:** 3
**Technical Novelty And Significance:** 2
**Empirical Novelty And Significance:** 2
**Recommendation:** 3

**Strength And Weaknesses:**

**Strength**

(1) it shows the traditional FL algorithms are unstable for non-IID data. (2) it proposes a new knowledge distillation algorithm Label-Driven Knowledge Distillation. (3) it proposes a new FL algorithm F2L.

**Weaknesses**

(1) what is the meaning of the first step of FL, direct LKD may get better results. (2) an additional dataset (Reference Datapool in Fig. 1) is introduced. Whether the distribution of this dataset is similar to the distribution of the teachers' training dataset. The comparison with other methods is not fair and the experimental results are not convincing. (3) lack of explanation of some symbols in the paper makes it less readable. (4) some lines in Fig. 1 are misplaced. (5) some references are missing. Many FL works have introduced KD. For example,
[1] Fine-tuning Global Model via Data-Free Knowledge Distillation for Non-IID Federated Learning. [2] FedBE: Making Bayesian Model Ensemble Applicable to Federated Learning. [3] FedMD: Heterogenous Federated Learning via Model Distillation. [4] Ensemble Distillation for Robust Model Fusion in Federated Learning.

**Summary Of The Paper:**

This paper proposes a FL framework named F2L. It combines hierarchical network design and knowledge distillation to solve the problem of non-IID of data. Its contributions are threefold:(1) it shows the traditional FL algorithms are unstable for non-IID data. (2) it proposes a new knowledge distillation algorithm Label-Driven Knowledge Distillation. (3) it proposes a new FL algorithm F2L.

**Summary Of The Review:**

See the Weaknesses.

---

### Official Review · Reviewer_3LcY · 2022-10-25

**Confidence:** 4
**Correctness:** 4
**Technical Novelty And Significance:** 3
**Empirical Novelty And Significance:** 3
**Recommendation:** 3

**Clarity, Quality, Novelty And Reproducibility:**

Some clarification questions:
●	Does the performance of F2L depend on the number of regions R?
●	Table 1 provides the results for two extreme values of alpha - 0.1 and 1. It would be helpful if the results are also shown on a non-extreme third value.
●	Could the system be extended to more than 2 levels of hierarchy? Would there be a benefit in doing that?
●	Do Figures 2a and 2b represent training on EMNIST? Fig 2b needs more description.


**Strength And Weaknesses:**

Strengths -
1.	They offer to solve the two problems of scalability and data heterogeneity simultaneously.
2.	Since a hierarchical model is able to train the sub-regions before doing a global aggregation, the proposed design is claimed to also achieve computational efficiency.
3.	The objective function for online distillation in Eq. (9) is well formulated that includes the goal of reducing the generalization gap between the regions and changes in the global model; this is so that the model does not forget the crucial characteristics of the old global model.
4.	Equations (11) and (12) recommend how to set the \lambda_2 and \lambda_3 in terms of \lambda_1. This reduces the number of hyperparameters to be chosen by the user.
5.	F2L is compatible and integrable with other FL techniques.

Weaknesses -
1.	The main drawback of the design is that it relies heavily on a centrally available dataset. One of the two primary goals of the system is to handle non-IIDness in the data, which raises the question - how does the performance of F2L depend on the quality of the root dataset at the server. How well does the root dataset represent the non-IIDness present among the clients? How is scalability affected if the root dataset is not updated to well represent the newly joined clients? More experiments are required to convince the reader that the system can do well even when the root dataset does not exactly represent the data distribution among the clients.
2.	Table 1 shows that F2L performs significantly better than Fed-Distill. The lower performance of the other benchmarks can be attributed to the fact that they do not leverage any information from a root dataset. What essentially leads to this improvement with respect to Fed-Distill? Do they both use the same root datasets? Is Fed-Distill well tuned for best performance?
3.	Figure 2c shows the performance of F2L when a client is injected into the system midway during the training. F2L can be seen to perform better than vanilla FL. Can this be attributed to knowledge distillation? How would it compare with Fed-Distill? How sensitive are the observations with respect to the knowledge distillation parameters - lambda and temperature?
4.	F2L relies on switching between LKD and FedAvg after sufficient convergence has happened. How is this threshold chosen? What can be a general way to choose this value for any dataset?
5.	Figure 3 shows that a student can outperform a teacher in F2L. This experiment was performed on EMNIST. Does this observation hold in general, independent of the dataset? If not, what conditions does this depend on?


**Summary Of The Paper:**

The paper presents a novel FL framework called Full-stack FL (F2L). It aims at solving two important problems in FL - scalability, and robustness in the presence of heterogeneous data. To solve the first problem, they propose a hierarchical design in FL where several smaller FL systems are connected via a global server. To handle heterogeneity, they propose a new label-driven knowledge distillation (LKD) technique at the global server which leverages the advantage of the hierarchy in the design to learn useful information from the sub-regions in the network to achieve fast convergence by reducing the generalization gap between regions.

**Summary Of The Review:**

Both hierarchical learning and KD exist in literature. The authors claim their KD method is novel though. The contributions are significant only if we have a well represented root dataset in my opinion, which may not be able to cope with scalability in practice, which is in fact the main contribution claim of the paper.

---

### Official Review · Reviewer_yujU · 2022-10-28

**Confidence:** 4
**Correctness:** 2
**Technical Novelty And Significance:** 2
**Empirical Novelty And Significance:** 2
**Recommendation:** 3

**Clarity, Quality, Novelty And Reproducibility:**

clarity: the paper is a bit hard to follow

quality: the experiments need to be improved

novelty: the hierarchy structure of FL is somehow new if the authors can really build such an FL system including hundreds of clients distributed in different regions (instead of only simulation in one computer)



**Strength And Weaknesses:**

S1. The proposal of a hierarchical structure for FL is reasonable.

W1. The paper is very hard to follow, as a lot of design considerations are proposed, but which parts are the most novel ones are unclear.

W2. The comparison with baselines is unfair. It seems that the authors assume there is some test data in the server so that every region’s model can be validated for each label; then label-driven distillation can be proposed. However, the baselines seem not to have this assumption and can work well without the centralized test data. In other words, the authors’ method uses external knowledge (i.e., server test data), then performance improvement is expected.

W3. In practice, it is very hard to collect test data for the server; even if the server can collect some, such data may be very biased (because many clients will not allow the server to collect data; this is actually why FL is needed). Then, the server’s validation accuracy may be doubtful. Hence, the application of the proposed mechanism in practice is doubtful.

W4. The experiment settings are manually controlled so whether the regional hierarchy can work in practice is still unknown. From my reading of the paper, I think the mechanism may work well when different regions have non-iid samples (the experimental setting); however, if non-iid happens for the clients in a region, the mechanism may still fail. The authors need to use realistic data/region partitions instead manually controlled ones to validate the usefulness of the proposed mechanism.

W5. While the hierarchy structure of FL is sensible, the authors may have real experiments to deploy hundreds of clients in different regions to verify the practicality of the hierarchy structure. Otherwise, how this structure is useful in practice is unclear.


**Summary Of The Paper:**

This paper proposes a hierarchical FL framework with a new label-driven distillation method to handle non-iid FL scenarios

**Summary Of The Review:**

In general, I think that the hierarchy structure has potential in real FL systems. Meanwhile, the challenge of building such a system may be more in other parts, e.g., communication/node joining & dropping, instead of gradient aggregation. For the label-driven distillation, I think the authors' comparison with baselines is a bit unfair, as the authors' mechanism has more information (i.e., test data in the server).

---

### Decision · Program_Chairs · 2023-01-20

**Decision:**

Reject

**Justification For Why Not Higher Score:**

No rebuttal, low scores, clear reject.

**Justification For Why Not Lower Score:**

N/A

**Metareview: Summary, Strengths And Weaknesses:**

Federated Learning (FL) meets two challenges when applied in real-world applications: scalability, particularly in massive IoT networks, and being robust against a heterogeneous data environment. To address the first challenge, the authors provide an FL framework named Full-stack FL (F2L). F2L uses a hierarchical network architecture, the authors claim that this makes it easy to extend the FL network without reconstructing the entire system. To address the second challenge, a new label-driven knowledge distillation (LKD) technique is proposed at the global server. The authors claim that unlike existing knowledge distillation techniques, LKD can train a student model that incorporates knowledge from all the teacher models. The authors also carryout various experiments  to show that F2L can potentially improve FL efficiency in all global distillations, and quickly converges as global distillations occur, instead of increasing on each communication cycle.

The reviewers raised a variety of technical concerns. The authors did not provide a rebuttal. Therefore, I recommend rejection.